# Small-Dose Sunitinib Modulates p53, Bcl-2, STAT3, and ERK1/2 Pathways and Protects against Adenine-Induced Nephrotoxicity

**DOI:** 10.3390/ph13110397

**Published:** 2020-11-17

**Authors:** Mohamed A. Saleh, Ahmed M. Awad, Tarek M. Ibrahim, Nashwa M. Abu-Elsaad

**Affiliations:** 1Department of Clinical Sciences, College of Medicine, University of Sharjah, Sharjah 27272, UAE; 2Department of Pharmacology and Toxicology, Faculty of Pharmacy, Mansoura University, Mansoura 33516, Egypt; ahmedawad@mans.edu.eg (A.M.A.); tarekmos@mans.edu.eg (T.M.I.); nashabuelsaad@mans.edu.eg (N.M.A.-E.)

**Keywords:** sunitinib, p53, nephrotoxicity, TGF-β1, ERK1/2, Bcl-2

## Abstract

The therapeutic use of numerous pharmacological agents may be limited due to their nephrotoxicity and associated kidney injury. The aim of our study is to test the hypothesis that the blockade of tyrosine kinase-linked receptors signaling protects against chemically induced nephrotoxicity. To test our hypothesis, we investigated sunitinib as an inhibitor for tyrosine kinase signaling for both vascular endothelial growth factor receptor (VEGFR) and platelet-derived growth factor receptors (PDGFR) against adenine-induced nephrotoxicity. Four groups of adult male Swiss albino mice were investigated: normal group, adenine group, sunitinib group, and the adenine+sunitinib group that received concurrent administration for both adenine and sunitinib. Kidney function and oxidative stress biomarkers were analyzed. Tubular injury and histopathological changes were examined. Renal expression of B-cell lymphoma-2 (Bcl-2), the tumor suppressor p53, transforming growth factor beta-1 (TGF-β1), phospho-extracellular signal-regulated kinase 1/2 (p-ERK1/2), and phospho-signal transducer and activator of transcription (phospho-STAT3) were measured. The results obtained showed significant improvement (*p <* 0.05) in kidney function and antioxidant biomarkers in the adenine+sunitinib group. Kidney fibrosis and tubular injury scores were significantly (*p <* 0.05) less in the adenine+sunitinib group and that of p53 expression as well. Furthermore, sunitinib decreased (*p <* 0.5) renal levels of TGF-β1, p-ERK1/2, and phospho-STAT3 while elevating Bcl-2 expression score. In conclusion, sunitinib diminished adenine-induced nephrotoxicity through interfering with profibrogenic pathways, activating anti-apoptotic mechanisms, and possessing potential antioxidant capabilities.

## 1. Introduction

End-stage renal disease (ESRD) is considered the direct sequelae of an unsuccessful regeneration of the insulted renal tissues. The concomitant progression of the renal failure in patients with chronic kidney diseases (CKD) is a function of the accompanying tubule-interstitial fibrosis, glomerular insufficiency, and tubular atrophy [1].

The infiltration of immune cells into the tubulointerstatium; activation of fibroblasts and increased extracellular matrix production and deposition; and massive disappearance of arterioles and capillaries (rarefaction) represents the crucial cellular changes that accompany CKD [2]. The current effective manipulation of the CKD problem is either dialysis or kidney graft. Each of these two solutions has its own problems. Dialysis cannot restore any of the lost kidney functions; meanwhile, kidney graft is mainly not available to the patient population in developing countries in addition to the usual shortage of donors. 

Tyrosine kinase inhibitors may represent a reasonable solution for the problem. Tyrosine kinases are considered one of the main players in regulation of the normal kidney functions via their role in cell signaling through the angiopoietin (Ang)-Tie signaling pathway [3]. The pathologic alteration of this pathway activity, which accompanies the inflammatory sequences of renal injury, leads to dysregulation of the normal kidney functions, carcinogenesis, vascular remodeling, and fibrogenesis [4,5]. Thus, tyrosine kinase inhibitors may represent a therapeutic option in CKD management guidelines.

It is well established now that CKD is accompanied by an increase in the production of receptor tyrosine kinase ligands and transforming growth factors with subsequent signaling processes leading to the final stage of kidney fibrosis [6]. Sunitinib is an orally administered receptor tyrosine kinase inhibitor that inhibits the signaling pathway of two main growth factor receptors, which are vascular endothelial growth factor receptors (VEGFR) and platelet-derived growth factor receptors (PDGFR).

Kidneys in animal models of diabetes that treated with monoclonal antibodies targeting VEGFR exhibited significant improvement in function with a reduced urinary albumin excretion rate [7]. Regarding PDGFR induced-renal fibrosis, it has been shown that PDGFR signaling blockade leads to an attenuation of proliferation and inhibiting the differentiation of pericytes into myofibroblasts in progressive kidney disease [8]. Finally, a dual inhibition of VEGFR and PDGFR via nintedanib leads to an inhibition of cell proliferation and diminishing collagen type I deposition in ex vivo human precision-cut kidney slices [9]. Our study was designed to evaluate the effect of sunitinib as a dual-targeted receptor tyrosine kinase inhibitor against adenine-induced nephrotoxicity with investigating its possible therapeutic targets to reverse the developed renal fibrosis.

## 2. Results

A significant increase in renal function biomarkers (creatinine, urea. and uric acid) in serum was observed in mice after receiving adenine for 4 weeks (*p <* 0.05). Urine albumin was also significantly increased (*p <* 0.001) by adenine. Receiving sunitinib alone had no significant effect on measured biomarkers when compared to the normal group. On the other hand, concurrent sunitinib administration with adenine decreased creatinine and urea (*p <* 0.001) levels in serum compared to receiving adenine alone (Figure 1a and Figure 1b, respectively). Serum uric acid level was significantly *p <* 0.05 decreased in the adenine+sunitinib group compared to adenine group (Figure 1c). Albumin concentration in urine was significantly (*p <* 0.001) elevated in the adenine+sunitinib group (Figure 1d).

The administration of adenine alone significantly (*p <* 0.001) decreased the total antioxidant capacity (Figure 2a) and catalase activity (Figure 2b) in renal tissue compared to the normal group. When adenine was combined with sunitinib, levels of these antioxidant biomarkers were significantly (*p <* 0.05) elevated compared to adenine alone but remained significantly less than normal group levels (*p <* 0.05). Similarly, a reduced glutathione (GSH) level was significantly (*p <* 0.001) higher in the adenine+sunitinib group than in the adenine group and lower than its level in the normal group (Figure 2c). An elevated level of MDA in the adenine group was significantly (*p <* 0.01) lowered when adenine was co-administered with sunitinib (Figure 2d).

A significant (*p <* 0.001) increase in p-extracellular signal-regulated kinase 1/2 (p-ERK1/2) expression in kidney homogenates was found in the adenine group (Figure 3a) when compared to the normal group. Sunitinib concurrent administration with adenine significantly decreased the expression of p-ERK1/2 levels (*p <* 0.001). Comparable results were obtained for phospho-signal transducer and activator of transcription (phospho-STAT3) levels (Figure 3b). In the adenine group, transforming growth factor beta-1 (TGF-β) levels were significantly (*p <* 0.001) high when compared to the normal group. In the adenine+sunitinib group, the fibrogenic marker level was significantly (*p <* 0.001) lowered compared to the adenine group but still higher (*p <* 0.05) than the normal group level (Figure 3c).

An examination of kidney sections revealed marked tubular injury and inflammatory cells recruitment (Figure 4a) in sections isolated from the mice that received adenine. Sunitinib significantly decreased the tubular injury score in the adenine+sunitinib group significantly (*p <* 0.001) compared to the adenine group (Table 1). Collagen deposition and fibrosis was markedly high in adenine group sections (Figure 4b). Fibrosis score was reduced in the adenine+sunitinib group by about 2-fold compared to the adenine group alone (Table 1).

The administration of adenine significantly (*p* < 0.001) increased the renal tissue level of both platelet-derived growth factor (PDGF) (Figure 5a) and vascular endothelial growth factor (VEGF) (Figure 5b). Sunitinib, when concurrently administered with adenine, significantly (*p* < 0.001) decreased both PDGF and VEGF renal levels to their normal levels. In addition, sunitinib restored PDGF renal levels its normal values, but renal VEGF levels were sill higher (*p* < 0.001) when compared to the corresponding normal values.

The expression of p53 in immuno-stained kidney sections is depicted in Figure 6a. Mice that received adenine showed a significantly high (*p <* 0.001) p53 expression score compared to the normal group score (≈3-fold). Sunitinib significantly decreased p53 renal expression (*p <* 0.01) when combined with adenine compared to adenine alone by ≈2.2-fold (Figure 6b). The p53 score in the sunitinib and adenine+sunitinib groups was significantly higher than the normal group score (*p <* 0.05, 0.001 respectively). 

Immuno-stained kidney sections showing an expression of the anti-apoptotic marker Bcl-2 are represented in Figure 7a. Expression was significantly (*p <* 0.01) decreased by about 1.8-fold decrease compared to the normal group. In the sunitinib+adenine group, Bcl2 expression was significantly increased (*p <* 0.001) compared to both the normal and adenine groups (Figure 7b).

Expression of the apoptotic marker caspase-3 was significantly (*p* < 0.001) increased in sections isolated from adenine group mice when compared with those of normal group sections (Table 1). Sunitinib administration concurrently with adenine decreased the percentage of caspase-3 positive cells significantly (*p* < 0.001) when compared to the adenine group percentage levels (Figure 8); however, this percentage remained significantly higher when compared to the normal group (*p* < 0.05).

## 3. Discussion

Chronic kidney diseases are considered an aggravated problem in all countries whether developed or underdeveloped. The current strategy for the medicinal treatment of CKD focuses on managing the complications of the disease rather than the disease itself, which makes the current guidelines highly complicated. Receptor tyrosine kinase inhibitors represent a rising hope for a new approach to manage the underlying mechanisms of kidney diseases progression.

Sunitinib is a receptor tyrosine kinase inhibitor that proved efficacy as a first-line therapy for metastatic renal cell carcinoma, especially in patients who cannot tolerate immunotherapy with monoclonal antibodies [10]. Sunitinib targets both PDGFRs and VEGFRs, inhibiting tumor angiogenesis. These two targeted molecules play a crucial role in deterioration of the renal functions that characterizes end-stage renal failure [11,12].

The results of the present work showed that tested parameters of renal function were improved by the administration of sunitinib. In line with biochemical results, histopathological changes in renal tissue including tubular injury and fibrosis were also improved. Notably, sunitinib decreased renal tissue levels of VEGF and PDGF when administered with adenine. These effects suggest the ability of sunitinib to interfere with the VEGF regulatory role on normal renal function. The upregulation of VEGF plays a role in the pathogenesis of renal dysfunction through stimulating mesangial cells proliferation and ECM deposition [13]. Furthermore, podocyte-derived VEGF is believed to participate in the glomerular capillary hyperpermeability and may lead to albuminuria and proteinuria associating glomerular hypertrophy [14]. Subsequently, the inhibition of VEGF represents a possible therapeutic target to ameliorate the structural and functional changes involved in CKD. In addition to VEGF, sunitinib can target and inhibit the PDGF/PDGFR system, which represents a key factor in the pathogenesis renal fibrosis [15].

An interplay between some intracellular communication lines, tyrosine kinases, and chronic renal injuries development is suggested. The concurrent administration of sunitinib with adenine restored the balance in the oxidative status as indicated by the measured antioxidant biomarkers. One of the important signaling pathways that link the increased oxidative stress to acute kidney injury is the mitogen-activated protein kinases (MAPKs) [16]. Extracellular signal-regulated kinase (Erk) 1/2, Erk5/BMK1, c-Jun N-terminal kinase (JNK), and p38-MAPK are the four different MAPK pathways in mammalian cells. They help kidney adaptation to the inflammatory infiltration-induced hypoxia by increasing the proliferation and deposition of the extracellular matrix. Such signaling pathway activation is stimulated by PDGFR and VEGFR coupling with their ligands, and the net result would be tissue fibrosis [17]. The results of our study reveal that sunitinib administration can inhibit the activity of p-ERK1/2 (MAPK). This is consistent with the effect of targeting VEFG/VEGFR and PDGF/PDGFR by sunitinib.

In various experimental and human nephropathies, signal transducer and activator of transcription (STAT3) activation has been reported in the injured kidney, showing a profibrogenic role [18]. There is a crosstalk between the activation of receptor tyrosine kinases and STAT pathway activation. They act as docking sites for STATs phosphorylation and activation [19]. The activation of STAT3 was found to be induced by PDGF, VEGF, and MAPK. In models of fibrosis, the increased TGF-β1 expression [20] and the subsequent collagen I production were attributed to STAT3 [21]. In the present work, decreasing the expression of phospho-STAT3 may account for the protection against the development of kidney fibrosis and in turn the progression of kidney diseases, as previously shown by [22,23].

Additionally, obtained data showed a reduced TGF-β1 production in renal tissue by sunitinib. A complex interplay between tyrosine kinase inhibitors and the fibrogenic role of TGF-β1 has been reported by various studies. Some studies postulated that VEGFR activation is critical for TGF-β1 overproduction and anti-VEGF can downregulate the expression of alpha smooth muscle actin (α-SMA) and TGF-β1 [24], whereas other studies reported that TGF-β1 induces VEGF and VEGFR overexpression in mouse dendritic cell lines under hypoxic conditions [25].

Our study suggests a cooperativity between the tumor suppressor p53 and tyrosine kinases in renal injury progression. Interestingly, p53 forms a homotetrameric transcription factor that directly regulates ≈500 target genes, thereby controlling a various range of cellular processes, including DNA repair, cell cycle arrest, metabolic adaptation, and apoptosis [26]. Increasing p53 phosphorylation can enhance renal damage in different models of renal injury, possibly by promoting tubular apoptosis and proliferation inhibition [27]. An interaction between Smad3 and the RTK/Ras/ERK1/2 cascade-phosphorylated N-terminus of p53 was also reported in vitro [28]. On the other hand, its inhibition was found to block the activation of STAT3 and collagen expression during TGF-β1 treatment [29]. Accordingly, either the pharmacological or genetic blockade of p53 can attenuate renal inflammation, apoptosis, and fibrosis. Likewise, sunitinib may halt fibrosis progression by decreasing renal p53 expression.

In addition to fibrogenic pathways, regulating apoptosis in renal tissue can help control CKD development. The anti-apoptotic B-cell lymphoma-2 (Bcl-2) prevents the activation of the apoptotic caspases in the mitochondria, leading to the retardation of cell death [30]. Jin et al. reported that receptor tyrosine kinase inhibitors increased Bcl-2 expression levels in cancer cell lines, and this was considered a resistance mechanism of cancer cells against tyrosine kinase inhibitors [31]. In kidney diseases, it was noted that Bcl-2 may be downregulated with subsequent enhancement in a Bax-induced apoptotic effect. Furthermore, it has been demonstrated that there was a decrease in Bcl-2 expression in unilateral urethral obstruction [30]. Herein, and as opposed to cancer treatment, sunitinib increased renal Bcl-2 levels and reduced those of caspase-3. Numerous studies reported a key role of caspase dependent apoptosis in the development of tubular injury and kidney dysfunction [32,33]. Thus, sunitinib possesses the potential effects to modulate apoptotic pathways and to halt renal tissue injury and subsequent fibrosis progression.

## 4. Materials and Methods

### 4.1. Study Design

Adult male Swiss albino mice (CD-1 strain, *n* = 40, 25–30 g) were purchased from VACSERA (Agouza, Giza, Egypt). Mice (10/cage) were kept for 2 weeks of accommodation with free access to food and water. The study was approved by the Scientific Research Ethics Committee, Faculty of Pharmacy, Mansoura University (approval number: 2020-22). The guidelines and ethical principles for the care and use of laboratory animals comply with the National Institutes of Health guide (NIH Publications No. 8023, revised 1978) for the care and use of laboratory animals.

After accommodation, food was restricted to the average consumed during accommodation (50 g/cage/day) in all groups along the study. Nephrotoxicity was induced by adding 0.2% *w*/*w* adenine (fine white powder, Oxford laboratory Pvt. Ltd., Mumbai, India; PubChem CID: 190) to food daily for 4 weeks [34,35].

The study included four groups (10 mice/group): normal group: received sunitinib vehicle (6 mL/kg/day); adenine group: received adenine as described above in addition to sunitinib vehicle; sunitinib group: received sunitinib (LC laboratories, Woburn, MA, USA; PubChem CID: 5329102) orally in a dose of 25 mg/kg/day as a suspension (1.3% *w*/*v* in normal saline) for 4 weeks [36]; adenine+sunitinib group: received both adenine and sunitinib as mentioned above. Table 2 shows a schematic presentation of the study’s design.

At the study end, mice were transferred to metabolic cages for 24 h to collect urine samples. Blood samples were collected (24 h after the last adenine dose) from anesthetized mice (70 mg/kg thiopental, i.p.) *via* cardiac puncture, centrifuged at 3000 rpm for 15 min at 4 °C, and serum was separated. The two kidneys were isolated, cut into pieces, and used to prepare kidney homogenate (10% *w*/*v* in ice-cooled phosphate buffer) and paraffin blocks.

### 4.2. Kidney Function Biomarkers

Serum creatinine, urea, and uric acid were assigned spectrophotometrically (at 546, 578, and 548 nm, respectively) using commercial kits (Biomed Diagnostics, Egy-Chem, Cairo, Egypt) according to the manufacturer’s instructions. Albumin was measured in urine (ABC Diagnostics, New Damietta, Egypt) spectrophotometrically at 590 nm.

### 4.3. Oxidative Stress and Antioxidant Biomarkers

Malondialdehyde (MDA) level as an indirect marker for lipid peroxidation was measured in homogenate as previously described by Ohkawa et al. [37]. Catalase activity in the kidney homogenate was determined using a method described by Aebi et al. [38]. Furthermore, renal levels of reduced glutathione (GSH) and total renal antioxidant capacity were measured using commercially available kits (Bio-diagnostic, Giza, Egypt) spectrophotometrically.

### 4.4. ELISA

Using the sandwich ELISA technique, renal tissue levels of phospho-STAT3, p-ERK1/2 (Assay Biotechnology Co., Fremont, CA, USA), TGF-β1 (Cloud Clone Corp., Katy, TX, USA), PDGF, and VEGF (Cusabio Biotech Co., Ltd., Houston, TX, USA) were measured according to the enclosed instructions. Absorbance was assigned at a wavelength of 450 nm using an automatic reader (model: DIAReader ELX808IU).

### 4.5. Tubular Injury and Interstitial Fibrosis

Tubular injury was scored in kidney sections (4 μm) stained with hematoxylin–eosin using the modified Kinomura et al. scoring system [39]. Briefly, twenty random fields were examined with a light microscope (BX51 Olympus optical microscope, Olympus cooperation, Tokyo, Japan) at a magnification of X400 and quantified for histopathological changes as swelling, desquamation from the tubular basement membrane, necrosis, and vacuolar degeneration. Scores from 0 to 5 were assigned according to the percentage of the tubular histopathological changes (0: normal; 1: <20% 2: 20–40%; 3: 40–60%; 4: 60–80%; and 5: 80–100%).

Interstitial fibrosis was quantified in sections stained with Masson’s trichrome using a color image analyzer (J 1.32) avoiding blood vessels [40].

### 4.6. Immunohistochemistry

Renal sections were deparaffinized by serial washing with different alcohol concentrations. Sections were hydrated and immersed in ethylenediaminetetraacetic acid (EDTA) solution for antigen retrieval (5 min in 1 mM solution with a pH 8 at 121 °C). Then, sections received treatment for 10 min with hydrogen peroxide (3% in methanol) to stop endogenous peroxidase activity and for protein block. Afterwards, sections were incubated with one of the following primary antibodies: anti-p53 polyclonal antibody (ab131442) (Abcam, Cambridge, United Kingdom), Bcl-2 polyclonal antibody (PA5-20068) (Thermo Fischer scientific, Rockford, IL, USA), and capase-3 polyclonal antibody (ab13847) (Abcam, Cambridge, United Kingdom). As a negative control procedure, normal mouse serum was used instead of the primary antibody. After incubation with primary antibodies, slides were rinsed using phosphate buffer saline three times and incubated with the secondary antibody anti-rabbit IgG provided by DAKO EnVision™+ System Horseradish Peroxidase (Dako North America, Inc., Carpinteria, CA, USA) at 28 °C for 30 min. The expression of p-53 (cytoplasmic and nuclear), Bcl-2 (cytoplasmic), and caspase-3 (nuclear) was visualized using diaminobenzidine kits (liquid diaminobenzidine–substrate chromogen system, Dako) and Mayer’s hematoxylin as the counterstain. The percentage of positive cells in 100 cells/each high definition field was detected using a light microscope (Olympus cooperation, Tokyo, Japan) and data were represented as the mean of 20 fields/group [41].

### 4.7. Statistical Analyses

The significance level was set to *p <* 0.05. Scores of Bcl-2 and p53 are expressed as median with interquartile range and analyzed in addition to tubular injury using the Kruskal–Wallis test by rank followed by Dunn’s multiple comparison test. The rest of the data are expressed as the mean ± standard error of the mean and were compared using a one-way analysis of variance (ANOVA), followed by a Tukey–Kramer multiple comparison test as the post-hoc test. GraphPad Prism V5.01 (GraphPad Software Inc., San Diego, CA, USA) was used to perform statistical analysis and the construction of figures.

## 5. Conclusions

The multitargeted tyrosine kinase inhibitor sunitinib can halt adenine-induced nephrotoxicity and fibrosis possibly through interfering with phospho-STAT3 and p-ERK1/2 signaling pathways. It also shows an antioxidant capability and anti-fibrogenic effect that is mediated by downregulating TGF-β1 production and p53 expression. Finally, sunitinib can regulate apoptosis and tissue cell death through upregulating the anti-apoptotic member Bcl-2.

## Figures and Tables

**Figure 1 pharmaceuticals-13-00397-f001:**
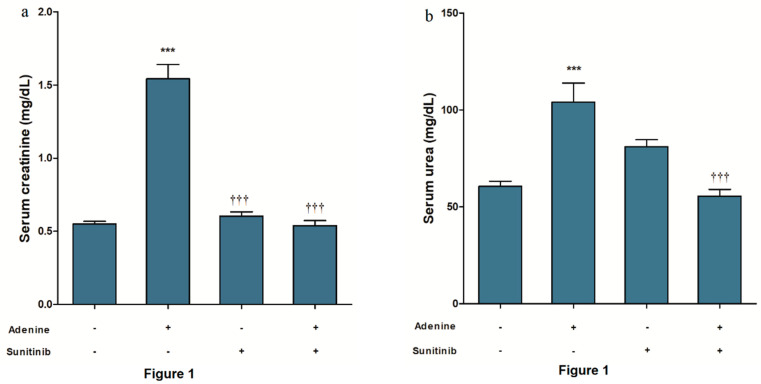
Sunitinib effect on level of serum (**a**), creatinine (**b**), urea (**c**), uric acid, and (**d**) urine albumin in an adenine model of nephrotoxicity (*n* = 8 mice/group). *, *** *p <* 0.05, 0.001 compared to control group; ^†^, ^†††^
*p <* 0.05, 0.001 compared to adenine group.

**Figure 2 pharmaceuticals-13-00397-f002:**
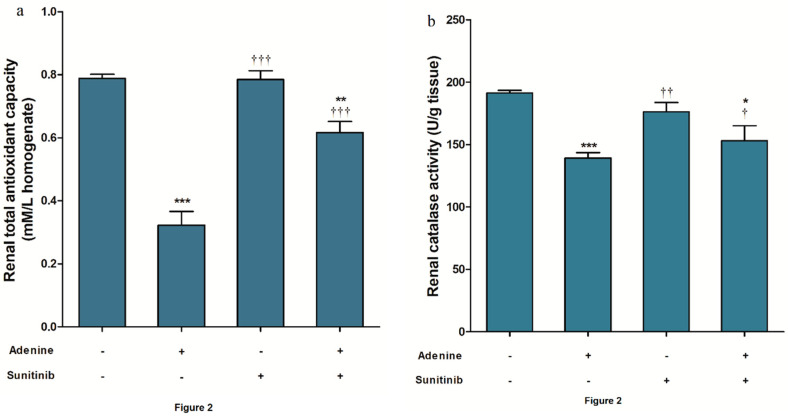
Sunitinib effect on renal tissue of (**a**) total antioxidant capacity, (**b**) catalase activity, (**c**) reduced glutathione (GSH) level, and (**d**) malondialdehyde (MDA) level in an adenine model of nephrotoxicity (*n* = 8 mice/group). *, **, *** *p <* 0.05, 0.01, 0.001 compared to normal group; ^†^, ^††^, ^†††^
*p <* 0.05, 0.001, 0.001 compared to adenine group.

**Figure 3 pharmaceuticals-13-00397-f003:**
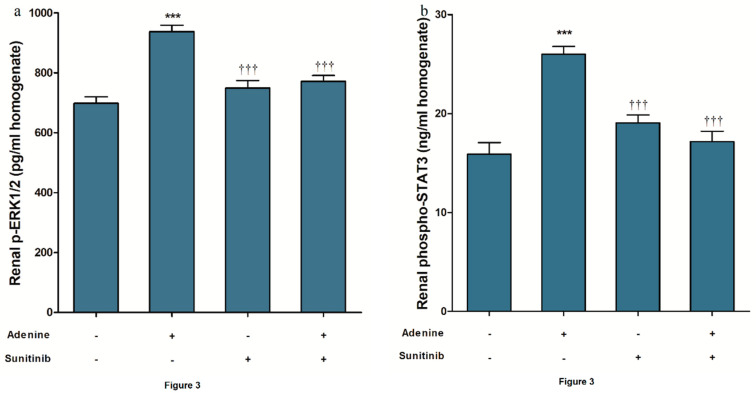
Sunitinib effect on renal tissue level of (**a**) p-extracellular signal-regulated kinase 1/2 (p-ERK1/2), (**b**) phospho-STAT3, and (**c**) transforming growth factor beta-1 (TGF-β1) in adenine model of nephrotoxicity (*n* = 8 mice/group). *, *** *p <* 0.05, 0.001 compared to normal group; ^†††^
*p <* 0.001 compared to adenine group.

**Figure 4 pharmaceuticals-13-00397-f004:**
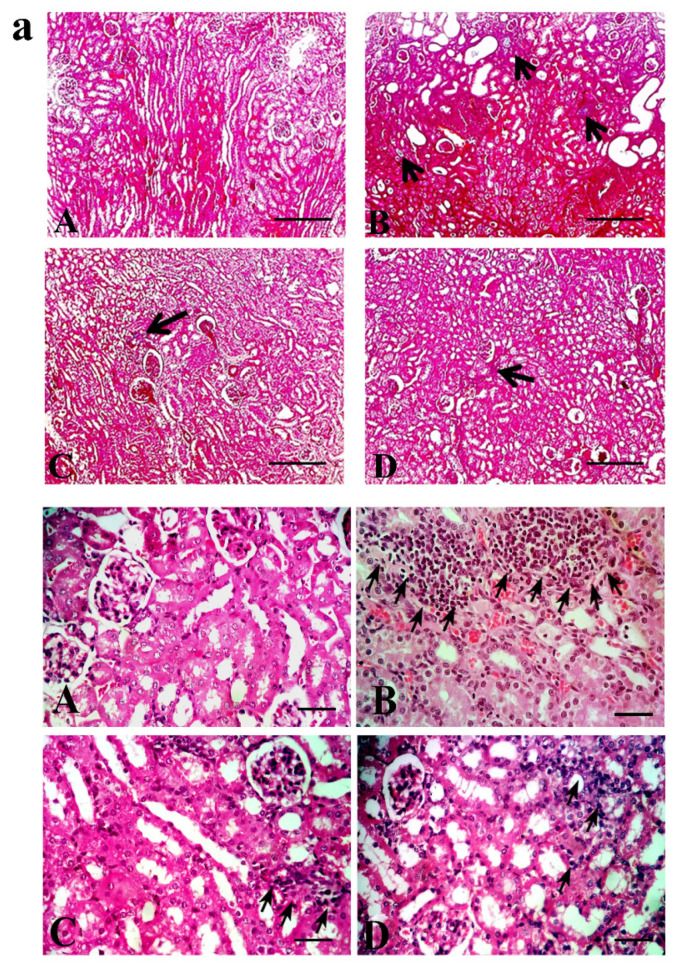
Representative kidney sections photographs stained with (**a**) hematoxylin–eosin stain: arrows indicate inflammatory cells infiltration, (**b**) Masson’s trichome stain: arrows indicate areas of fibrosis and collage deposition. (**A**) normal group, (**B**) adenine group, (**C**) sunitinib group, (**D**) adenine+sunitinib group. Low magnification (**upper panel**) X: 100 bar 100 and high magnification (**lower panel**) X: 400 bar 50.

**Figure 5 pharmaceuticals-13-00397-f005:**
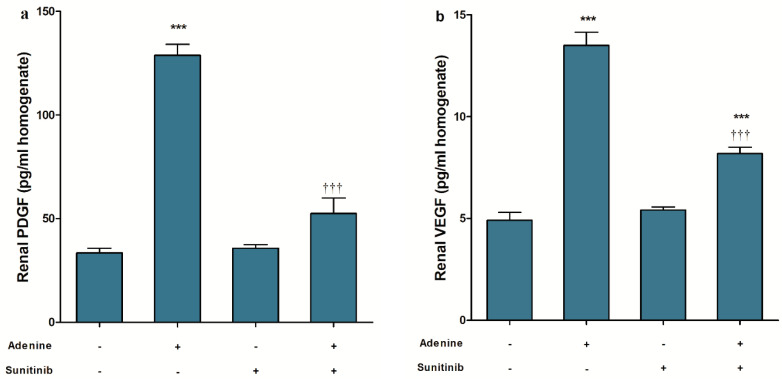
Effect of sunitinib on renal tissue levels of (**a**) platelet-derived growth factor (PDGF) (**b**) vascular endothelial growth factor (VEGF) in an adenine model of nephrotoxicity (*n* = 8 mice/group). *** *p* < 0.001 compared to control group; ^†††^
*p* < 0.001 compared to adenine group.

**Figure 6 pharmaceuticals-13-00397-f006:**
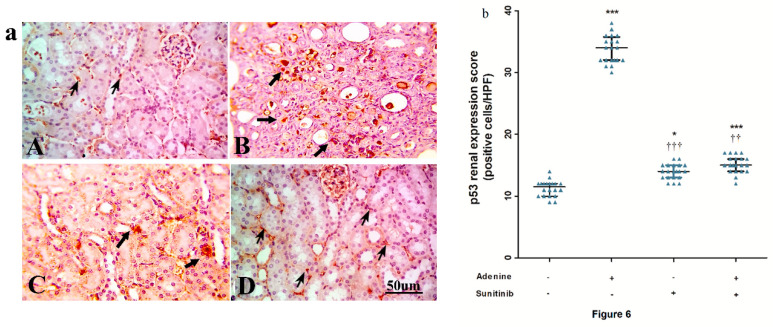
(**a**) Representative immuno-stained kidney sections photographs showing an expression of p53 (arrows) isolated from (**A**) normal group, (**B**) adenine group, (**C**) sunitinib group, and (**D**) adenine+sunitinib group. (**b**) Effect of sunitinib on kidney expression scores of tumor protein p53 in an adenine model of nephrotoxicity (*n* = 8 mice/group). Significance: *, *** *p <* 0.05, 0.001 compared to normal group; ^††^, ^†††^
*p <* 0.01, 0.001 compared to adenine group.

**Figure 7 pharmaceuticals-13-00397-f007:**
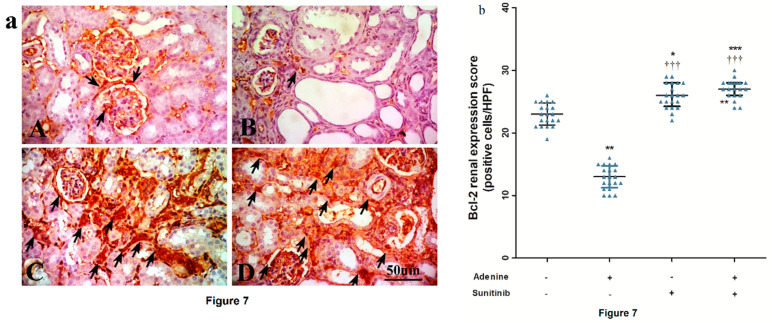
(**a**) Representative immuno-stained kidney sections photographs showing an expression of B-cell lymphoma (Bcl)-2 (arrows) isolated from the (**A**) normal group, (**B**) adenine group, (**C**) sunitinib group, and (**D**) adenine+sunitinib group. (**b**) Effect of sunitinib on kidney expression scores of Bcl-2 in an adenine model of nephrotoxicity (*n* = 8 mice/group). Significance: *, **, *** *p <* 0.05, 0.01, and 0.001 compared to the normal group; ^†††^
*p <* 0.01, 0.001 compared to the adenine group.

**Figure 8 pharmaceuticals-13-00397-f008:**
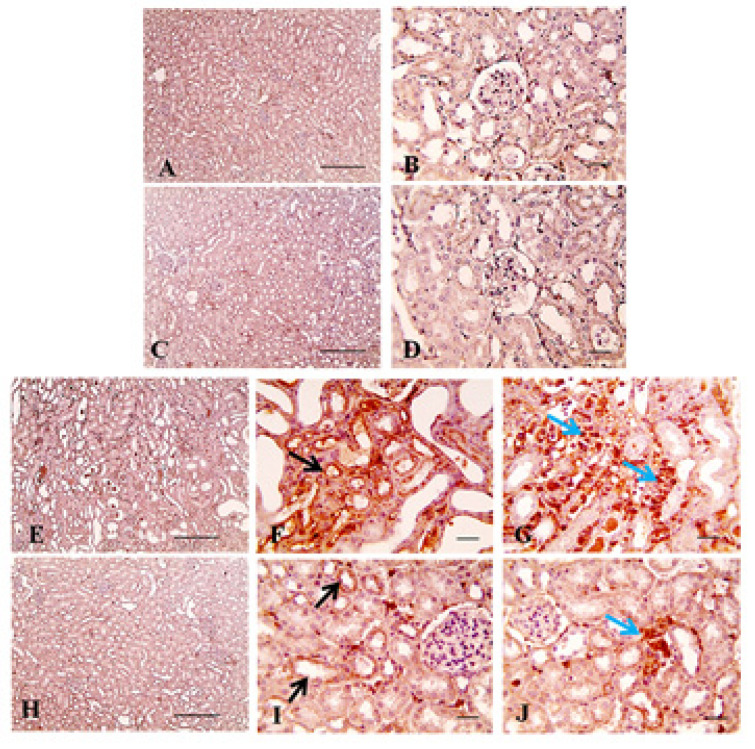
Representative immunostained kidney sections photographs (imuunohistochemistry counterstained with Mayer’s hematoxylin) showing expression of caspase-3 (arrows) isolated from normal group (**A**,**B**), sunitinib group (**C**,**D**), adenine group (**E**–**G**) and adenine+sunitinib group (**H**–**J**). Normal and sunitinib groups sections show negative caspase-3 staining. Renal sections from adenine group show strong positive brown staining in renal tubular epithelium (black arrow in **F**) and interstitial cells (blue arrows in **G**). Renal sections from the group received adenine+sunitinib show markedly decreased positive brown staining in renal tubular epithelium (black arrow in **I**) and interstitial cells (blue arrows in **J**) compared to adenine group. Low magnification X: 100 bar 100 (**A**,**C**,**E**, **H**) and high magnification X: 400 bar 50 (**B**,**D**,**F**,**G**,**I**,**J**).

**Table 1 pharmaceuticals-13-00397-t001:** Effect of sunitinib (25 mg/kg) on histopathological scores of renal tubular injury and interstitial fibrosis (/20 fields) and caspase-3 expression (positive cells percentage) in adenine induced-nephropathy in mice.

Treatment Groups (*n* = 10 Per Group)	Tubular Injury	Fibrosis Score	Caspase-3
normal	0.00 ± 0.000	1.14 ± 0.034	3 ± 0.611
adenine	2.35 ± 0.150	3.32 ± 0.094 ***	58.14 ± 4.444 ***
sunitinib	1.45 ± 0.114 ^†††^	1.57 ± 0.110 **^†††^	3 ± 0.583 ^†††^
adenine+sunitinib	1.65 ± 0.131 ^††^	1.52 ± 0.068 **^†††^	13.14 ± 1.334 **^†††^

**, *** *p <* 0.01, 0.001 respectively when compared with normal group. ^††,^
^†††^
*p <* 0.01, 0.001 respectively when compared with adenine group.

**Table 2 pharmaceuticals-13-00397-t002:** Schematic presentation of study design.

Treatment Groups (*n* = 10 Per Group)	Week				
	1	2	3	4	
normal	□	□	□	□	Sacrifice
adenine	☼□	☼□	☼□	☼□	
sunitinib	■	■	■	■	
adenine+sunitinib	☼■	☼■	☼■	☼■	

□ 0.9% *w*/*v* saline (6 mL/kg/day, by oral gavage). ☼ 0.2% w/w adenine (/day with food). ■ 25 mg/kg sunitinib suspension (1.3% *w*/*v* in saline, 6 mL/kg/day, by oral gavage).

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
