# Peer review of "Small-Dose Sunitinib Modulates p53, Bcl-2, STAT3, and ERK1/2 Pathways and Protects against Adenine-Induced Nephrotoxicity"

_pharmaceuticals, 2020, doi:10.3390/ph13110397_

Round 1

Reviewer 1 Report

Major points:

  1. The author mentioned how Sunitinib interplay with VEGF, PDGF or PDGFR, however, no evidence provided how VEGF or VEGFR or PDGF been regulated with the treatment in this study.
  2. The author found the upregulation of Bcl-2 in the tissue with sunitinib plus adenine, which is not enough to convince the prevention of apoptosis.
  3. The ELISA of p-STAT3 and p-ERK1/2 should normalize to total STAT-3 and ERK1/2 to show whether the phosphorylation of two proteins is not biased.

Minor points

  1. Figure legend not complete. Lack of information.
  2. Figure 4 and 5 magnification can be smaller
  3. Figure 6 seems false positive to me as the positive staining are 100% covered in B and C
  4. More detailed should be provided in the methods, especially the section of ELISA.

Author Response

Reviewer#1

Thank you for your valuable review and suggestions to improve our work. Kindly find the following responses to your comments.

Major points:

Comment#1: The author mentioned how Sunitinib interplay with VEGF, PDGF or PDGFR, however, no evidence provided how VEGF or VEGFR or PDGF been regulated with the treatment in this study.

Response#1: We measured renal tissue level of PDGF and VEDF using ELISA techniques as an added work. You can check the obtained results in the revised manuscript results section page (3) line (108) [highlighted in yellow and represented in figures (5a and 5b respectively)

Comment#2: The author found the upregulation of Bcl-2 in the tissue with sunitinib plus adenine, which is not enough to convince the prevention of apoptosis.

Response#2: We measured renal expression of apoptotic marker caspase-3 using immunohistochemistry technique as an added work. You can find obtained results in the revised version results section page (4) line (138) [highlighted in yellow]. Data are represented in table (1) [highlighted in yellow], represented by photographs figure (8) and discussed in discussion section page (5) line (219).

Comment#3: The ELISA of p-STAT3 and p-ERK1/2 should normalize to total STAT-3 and ERK1/2 to show whether the phosphorylation of two proteins is not biased.

Response#3: Unfortunately, we are not able to measure total STAT-3 and ERK1/2 as their kits are not available now for us. We will consider your recommendation in future work.

Minor points

Comment#1: Figure legend not complete. Lack of information.

Response#1: We revised all figure legends and improve them as appropriate, but it is not clear which figure legend is not complete?

Comment#2: Figure 4 and 5 magnification can be smaller

Response#2: You can find now representing photographs with smaller magnification (X100) in the revised version figure 4a upper panel and figure 4b upper panel

Comment#3: Figure 6 seems false positive to me as the positive staining are 100% covered in B and C

Response#3: We changed photographs B and C with more representable photographs in the revised manuscript. Kindly find the new photographs in figure (6a)

Comment#4: More detailed should be provided in the methods, especially the section of ELISA.

Response#4: Methods are represented according to the journal instructions. Only modifications in well-established methods are mentioned briefly to avoid repetition. We improved methods as possible in the revised version and highlighted in yellow in methods section. For ELISA section, we followed instructions of the manufacturer without changes and supplied the manufacturer details. Unfortunately, more details would increase similarity index.

Authors really appreciate your efforts

Reviewer 2 Report

The papers demonstrates therapeutic potential of sunitinib.

Although the paper is interesting, somme issues are highlighted and neet to be solved before this manuscript is published. 

  1. on what basis did the Authors choose the doses of the drug tested?
  2. what were the premises for conducting the experiment for the period of 4 weeks an no longer? Did the Authors posses any information regarding possible changes of the parameters and markers during the period of the study; maybe at the time point of 2 weeks from administration as well as at the first day (as a control of the control)?
  3. what was the dimension of kidney slices? Were they similar, especially in case of the material for homogenation. Please provide these informations.
  4. the Authors tested the level of albumin in urine samples. The most reliable method in order to indicate any kidney failure is, however, the urine albumin-to-creatinine ratio (UACR). Therefore, my question is whether the Authors could provide such informations.
  5. Please provide hte name of the company for light microscope
  6. the Authors provided with information that adenine/sunitinib group received these drug concurrently. Could you please describe the way of the administration, as adenine was given with food while sunitinib similarly per os. was in this case (for this  group) both drugs were mixed ? if yes, what was the total concentration of the mixture? Both drugs possibly have a different pharmacokinetic profile, therefore if the drugs were given simultanously, as described by the Authors, is the possibility exist that the observed effect is induced more by the first drug given than the second one? 
  7. the Authors provide data regarding changes in renal parameters, however there is no evidence that this beneficial effect of sunitinib is strictly related with is inhibitory activity towardstyrosinase kinase receptor
    1.  

Author Response

Reviewer#2

Thank you for your valuable review and suggestions to improve our work. Kindly find the following responses to your comments.

Comment#1: on what basis did the Authors choose the doses of the drug tested?

Response#1: Sunitinib dose was selected in the light of our preliminary studies and in the light of Tan et al., 2013 study. You can find the dose reference in Study design section page (6) line (246) highlighted in red.

Comment#2: what were the premises for conducting the experiment for the period of 4 weeks an no longer? Did the Authors possess any information regarding possible changes of the parameters and markers during the period of the study; maybe at the time point of 2 weeks from administration as well as at the first day (as a control of the control)?

Response#2:

  • The selected adenine model (dose and period) for inducing kidney injury and fibrosis is a previously established model [Nemmar, A., et al., 2016. Cell Physiol. Biochem. 38, 1703-1713; Diwan V, et al. Journal of pharmacological and toxicological methods. 2013 Sep 1;68(2):197-207]

Dose and period of adenine reference is highlighted in red Materials section.

  • Our study emphasizes mainly on chronic kidney damage and effect of tested drug on fibrosis. Before starting the study, we carried out preliminary trials and renal tissue and serum samples were obtained after administration of adenine at different time intervals (2, 4, 6 weeks). Significant changes in serum kidney function biomarkers in addition to histopathological changes and marked fibrosis were obtained after 4 and 6 weeks. Mortality rate after 6 weeks was significantly higher compared to 4 weeks possibly due to cardiovascular toxicity. For these reasons, the study was conducted for 4 weeks

Comment#3: what was the dimension of kidney slices? Were they similar, especially in case of the material for homogenization? Please provide this information.

Response#3: The homogenate was prepared by grinding 100 mg wet kidney slice in 1 ml ice cooled buffer. The concentration of the homogenate (10% w/v) is highlighted in red methods section. Kidney sections used for histopathological examination is 4 μm [highlighted in red]

Comment#4: the Authors tested the level of albumin in urine samples. The most reliable method in order to indicate any kidney failure is, however, the urine albumin-to-creatinine ratio (UACR). Therefore, my question is whether the Authors could provide such informations.

Response#4: Unfortunately, urine samples are not available now to measure UACR. We will consider this recommendation in our future work as it has a valuable significance.

Comment#5: Please provide hte name of the company for light microscope

Response#5: Microscope type: BX51 Olympus optical microscope (Olympus Corporation, Tokyo, Japan), you can find it now in the revised version highlighted in red in the method section.

Comment#6: the Authors provided with information that adenine/sunitinib group received these drug concurrently. Could you please describe the way of the administration, as adenine was given with food while sunitinib similarly per os. was in this case (for this group) both drugs were mixed? if yes, what was the total concentration of the mixture? Both drugs possibly have a different pharmacokinetic profile, therefore if the drugs were given simultaneously, as described by the Authors, is the possibility exist that the observed effect is induced more by the first drug given than the second one? 

Response#6:

  • adenine/sunitinib group received adenine in food as described in 4.1. materials section highlighted in red (by adding 0.2% w/w adenine ……….to food daily) in addition the group received sunitinib as described in 4.2. study design section highlighted in red (sunitinib group: received sunitinib ………… as a suspension (1.3% w/v in normal saline) orally by oral tube).
  • Adenine and sunitinib were not mixed.
  • There is no possibility that the observed effect is induced more by the first drug given than the second one. Adenine in the study is used to induce renal injury and fibrosis not as a treating drug.

Comment#7: the Authors provide data regarding changes in renal parameters, however there is no evidence that this beneficial effect of sunitinib is strictly related with is inhibitory activity towards tyrosinase kinase receptor

Response#7: We measured renal tissue level of the tyrosine kinase receptor ligands VEDF and PDGF using ELISA techniques as an added work. You can check the obtained results in the revised manuscript results section page [highlighted in yellow and represented in figures (5a and 5b). A significant (p<0.05) decrease in both VEDF and PDGF was observed in sunitinib treated group.

Authors really appreciate your efforts

Round 2

Reviewer 1 Report

The author has addressed my comments